# Dynamic imaging of crystalline defects in lithium-manganese oxide electrodes during electrochemical activation to high voltage

Qianqian Li[1,2,3], Zhenpeng Yao [3,5], Eungje Lee [4], Yaobin Xu [3], Michael M. Thackeray[4], Chris Wolverton[3], Vinayak P. Dravid[3] & Jinsong Wu[1,3]

Crystalline defects are commonly generated in lithium-metal-oxide electrodes during cycling of lithium-ion batteries. Their role in electrochemical reactions is not yet fully understood because, until recently, there has not been an effective operando technique to image dynamic processes at the atomic level. In this study, two types of defects were monitored dynamically during delithiation and concomitant oxidation of oxygen ions by using in situ high-resolution transmission electron microscopy supported by density functional theory calculations. One stacking fault with a fault vector $b/6[110]$ and low mobility contributes minimally to oxygen release from the structure. In contrast, dissociated dislocations with Burgers vector of $c/2[001]$ have high gliding and transverse mobility; they lead to the formation, transport and release subsequently of oxygen related species at the surface of the electrode particles. This work advances the scientific understanding of how oxygen participates and the structural response during the activation process at high potentials.

[1] State Key Laboratory of Advanced Technology for Materials Synthesis and Processing, Wuhan University of Technology, Wuhan, Hubei 430070, China. [2] Materials Genome Institute, Shanghai University, Shanghai 200444, China. [3] Department of Materials Science and Engineering, NUANCE Center, Northwestern University, Evanston, IL 60208, USA. [4] Chemical Sciences and Engineering Division, Argonne National Laboratory, Argonne, IL 60439, USA. [5] Present address: Department of Chemistry and Chemical Biology, Harvard University, 12 Oxford Street, Cambridge, MA 02138, USA. These authors contributed equally: Qianqian Li, Zhenpeng Yao. Correspondence and requests for materials should be addressed to Q.L. (email: qianqianli@shu.edu.cn) or to V.P.D. (email: v-dravid@northwestern.edu) or to J.W. (email: wujs@whut.edu.cn)

L ithium-ion batteries are today's dominant electrical energy storage technology; they continue to attract research and development support to improve their specific energy, power, durability, cycling stability, and safety for emerging markets such as electric vehicles[1,2]. Conventional cathode materials are typically lithium transition metal oxides and phosphates, such as $LiCoO_2$ (LCO)[3], $LiNi_{1-x-y}Mn_xCo_yO_2$ (NMC)[4], $LiMn_2O_4$ (LMO)[5], and $LiFePO_4$ (LFP)[6], that operate typically by (de-)intercalation of lithium during charge and discharge when the transition metal ions are oxidized and reduced, respectively, to store and release electrical energy. In this case, the specific capacity of the cathode, and hence the energy of the cell, is limited by the number of electrons per transition metal ion that can participate in the redox reactions. Lithium-rich metal-oxide electrodes that operate by both cationic (e.g., $Mn^{3+} \leftrightarrow Mn^{4+}$) and anionic (e.g., $O^{2-} \leftrightarrow O^{1-}$) or hybridized redox reactions are attractive materials because they have the potential to increase the energy storage capacity of lithium-ion batteries. Examples of materials that operate by anionic electrochemical reactions are $Li_2Ru_{1-y}Sn_yO_3$[7,8], $Li_3NbO_4$[9], $Li_3IrO_4$[10], $Li_5FeO_4$[11], $Li_2Mn_{1-y}M_yO_2F$[12], $Li_4Mn_2O_5$[13], and $Li_4(Mn,M)_2O_5$[14]. The reversible capacity of these reactions in lithium-rich materials is enabled by highly covalent metal-oxygen bonding[10] or by non-bonding oxygen $p$ orbitals generated by local lithium-excess configurations around O in the structure[11,14–16].

$Li_2MnO_3$ has a theoretical capacity of 459 mAh/g, which corresponds to the extraction of 2 Li per formula unit, when $Li_2MnO_3$ is activated chemically with acid[17] or electrochemically above 4.5 V vs. $Li^+/Li$ in lithium cells[18]. Lithium extraction, hydrogen-ion exchange, and oxygen loss reactions trigger a conversion of the parent layered structure to one with spinel-like features, which severely compromises the practical capacity, electrochemical potential, and cycling stability of the electrode and cell[17–21]. However, when integrated with a $LiMO_2$ component, the resulting $xLi_2MnO_3 \cdot (1-x)LiMO_2$ composite structures deliver a rechargeable capacity of more than 250 mAh/g after electrochemical activation of the $Li_2MnO_3$ component above 4.5 V[22]. Unfortunately, structural instabilities and voltage fade of these high capacity electrodes during cycling have thus far precluded their use in commercial lithium-ion battery products[23–29]. Although pure $Li_2MnO_3$ is now viewed as an unrealistic cathode material for commercial lithium-ion battery applications due to its rapid degradation, the underlying mechanism of the failure is unclear. Here we reported experimental finding and theoretical modeling results, which provides deeper insights on the underlying failure mechanisms.

Despite the progress made, a comprehensive understanding of the complex reaction mechanisms that occur during the electrochemical activation of structurally integrated $xLi_2MnO_3 \cdot (1-x)LiMO_2$ electrodes is still lacking. Such knowledge is critical if the limitations of anionic reactions are to be overcome. For this reason, in situ transmission electron microscopy (TEM) images of a $Li_2MnO_3$ electrode were recorded to monitor the dynamic structural changes that occur during the initial charge of the cell. A particular objective was to search for clues that might unravel the mechanism by which oxygen is lost from the $Li_2MnO_3$ electrode structure, while maintaining the tetravalent oxidation state of the manganese ions according to a simplified, ideal anodic electrochemical reaction[22,30–33]:

$$Li_2MnO_3 \rightarrow MnO_2 + 2\,Li^+ + \frac{1}{2}\,O_2 + 2e^-. \qquad (1)$$

Structural changes and oxygen loss that occur during delithiation of $Li_2MnO_3$ have already been reported by several groups, for example, by Rana et al.[30] and Yu et al[34]. These studies disclose, without specifying a mechanism, that delithiation occurs concurrently from both the lithium layer and the transition metal layer of the $Li_2MnO_3$ structure with the speculation that oxygen diffusion occurs sluggishly throughout the charged $Li_2MnO_3$ structure before $O_2$ gas is released at the particle surface[35–37].

Although defects are commonly observed in electrochemically cycled lithium-metal-oxide electrodes[38,39], they are often not mentioned when describing reaction mechanisms[40]. While it is still not clear if there is a connection between crystalline defects and oxygen redox and evolution reactions, defects induced into electrochemically cycled $Li_2MnO_3$ electrodes have been widely observed[38,39]. Stacking faults in the Mn-rich layers have been detected through X-ray diffraction and TEM measurements in both pristine $Li_2MnO_3$[34,41,42] and in partially delithiated "$Li_{2-x}MnO_3$" samples[39]. Other crystallographic defects, such as partial dislocations, have also been identified during the charging of $Li_2MnO_3$[40]. While these planar defects are generated to release mechanical strain and stress, their contribution to electrical energy storage and oxygen release in lithium-ion batteries remains unclear.

In this study, the relationship between crystalline defects and lithium extraction and oxygen evolution reactions in $Li_2MnO_3$ has been probed in detail. In situ TEM combined with density functional theory (DFT) calculations have been used to study the structural evolution of a $Li_2MnO_3$ electrode during the first charge (delithiation) and the mechanism of oxygen loss. The in situ TEM complemented by DFT calculation approach has proved to be an effective method for observing and analyzing the dynamic evolution of microstructure in battery electrodes during lithiation/delithiation cycles[43–46]. First, it allowed us to identify dynamic defects that appear in the $Li_2MnO_3$ structure during the electrochemical reaction, which are different to those that exist in the pristine state. Second, the results shed light on the reversibility of oxygen redox reactions at the atomic scale and the irreversibility of reactions that are associated with oxygen loss, which have significant implications for lowering the cycling efficiency of the electrode, particularly on the first cycle. Given the nature of electrochemical lithium extraction reactions, we presume that these dynamically formed defects result from changes in localized lithium-ion concentration. Two types of defects were observed: One is a stacking defect with a fault vector of $b/6[110]$, which has low activation energy for mobility that we tentatively associate with a reversible oxygen redox reaction (i.e., without oxygen loss). The second is a dissociated dislocation with Burgers vector of $c/2[001]$ that prompts the formation and release of $O_2$ at high electrochemical potentials (above 4.5 V), thereby contributing to capacity loss during the initial charge/discharge cycle. These discoveries and observations have possible implications for designing new materials and controlling reversible oxygen redox reactions in high capacity lithium-metal-oxide electrodes, notably those containing a $Li_2MnO_3$ component.

## Results

**As synthesized $Li_2MnO_3$ and its defects**. $Li_2MnO_3$ has a layered monoclinic structure (space group $C2/m$), with an atomic configuration, $Li[Li_{1/3}Mn_{2/3}]O_2$, in which layers of lithium (Li) alternate with layers of lithium and manganese ($Li_{1/3}Mn_{2/3}$) wherein the Li:Mn ratio is 1:2. (Supplementary Fig. 1). In the manganese-rich layer, the Li and Mn ions are arranged in a honeycomb fashion as illustrated in Supplementary Fig. 2a. Varying the stacking order of the manganese-rich ($Li_{1/3}Mn_{2/3}$) layers influences the crystal symmetry of the system. For example, as shown in the Supplementary Fig. 2b, the addition of a second manganese-rich layer generates "close-packed" AB stacking (note: with reference to the metal cation layers only)[47]. The

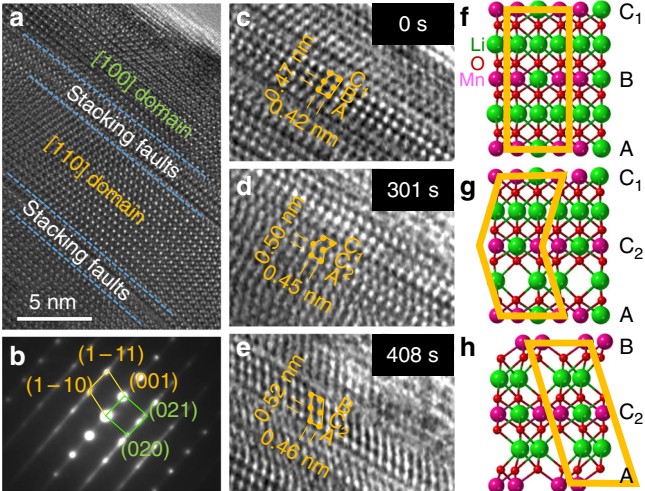

**Fig. 1** The exitance form and motion of stacking faults in lithium extraction process. **a** Transmission electron microscopy (TEM) image of (001) stacking faults in pristine $Li_2MnO_3$ with fault vector of $b/6[110]$, which are confirmed by a corresponding electron diffraction (**b**). During delithiation, the gliding of the $b/6[110]$ partial dislocation shears the stacking of the (001) plane from $ABC_1$ in pristine $Li_2MnO_3$ (**c**), to $AC_2C_1$ (**d**), and $AC_2B$ (**e**) after 0, 301, and 408 s, respectively; corresponding density functional theory (DFT) structural models are provided of pristine $Li_2MnO_3$ (**f**) and generated defects (**g**, **h**), respectively. The scale bar is 5 nm

addition of the third layer[41] would provide $ABC_1$ and $ABC_2$ orderings, corresponding to two basic stacking sequences: $C2/m$ ($ABC_1$) and $P3_112$ ($ABC_2$). More complex stacking orderings (e.g., well-ordered $C2/c$[48] and other faulting arrangements[39]) can be achieved by combining these two basic sequences in different ways.

Stacking faults exist in pristine $Li_2MnO_3$ samples when synthesized at 800 °C in air[39,40,42,49]. These planar defects can clearly be seen at the domain boundaries of two orientation variants, namely the [100] and [110] domains in Fig. 1a; the stacking fault disorder in pristine $Li_2MnO_3$ is confirmed by the offset and streaking of the diffraction spots in the corresponding TEM diffraction pattern shown in Fig. 1b. The defect shown in Fig. 1b is the result of a shear of the (001) layers; this defect can be described alternatively as a stacking fault bounded by a partial dislocation with a Burgers vector $b/6[110]$[50]. These planar defects release and accommodate strain and stress in the $Li_2MnO_3$ crystals. Similar defects have been observed by others in partially delithiated and relithiated samples[39], implying that these defects are active sites during charge and discharge reactions but might not participate significantly in oxygen loss reactions.

**Dynamic defects generated in delithiation.** In situ TEM images recorded during the initial stages of electrochemical delithiation, that is, after 0, 301, and 408 s, are depicted in Fig. 1c–e. A description of the cell design, which we have used effectively in previous studies of lithium insertion electrodes, such as $Co_3O_4$[45], is provided in detail in the Supplementary Information section. The structural changes that occur by the glide of the $b/6[110]$ dislocation during delithiation (Fig. 1c–e) were interpreted with the aid of structural models predicted by DFT calculations (Fig. 1f–h). The data show that, on gliding, the stacking sequence of a (001) lattice plane in a [100] domain changes from $ABC_1$ in pristine $Li_2MnO_3$ to an intermediate $AC_2C_1$ arrangement and subsequently to $AC_2B$. During this process, the (001) lattice spacing increases from ~0.47 to ~0.52 nm. The DFT models show, as expected, that the stacking fault defects are induced by

lithium-ion deficiencies and resulting crystal strain, making it energetically favorable for the (001) lattice planes to glide during the early stages of delithiation (Fig. 1f–h).

The in situ TEM studies revealed another defect type, not observed in the pristine $Li_2MnO_3$ electrode, but uniquely generated by the delithiation process. It can be described as a dissociated partial dislocation with Burgers vector $c/2[001]$ with a simultaneous transverse movement or "climbing" of the partial dislocation. More precisely speaking, the defect is a dissociated dislocation consisting of an antiphase boundary (with fault vector of $1/2[001]$) and the partial dislocation bounded to the antiphase boundary. The "fault plane" of the antiphase boundary is in the (100) plane with the atomic structure of the defect shown in Fig. 2b. Climbing of the dislocation refers to the movement of the defect across the (100) plane, while gliding refers to the movement in the (100) plane. This information leads us to believe that this active defect motion is largely responsible for the transport of an oxidized oxygen species within the $Li_2MnO_3$ crystal and the ultimate release of oxygen gas at the surface. Experimental (TEM) and computational evidence for this hypothesis is provided in Figs. 2–4.

As shown in the TEM image (Fig. 2a) and the computer model of a slightly delithiated $Li_{2-x}MnO_{3-\delta}$ structure (Fig. 2b), in which $x$ and $\delta$ are both small, defects exist as dissociated dislocations, or stacking faults, bounded by two partial dislocations. As the $c/2[001]$ Burgers displacement vector is perpendicular to the (001) lattice plane, the defect cuts the $Li_2MnO_3$ crystal into small fractions along the (001) plane. When further lithium ions are electrochemically removed from the structure, the density of defects increases significantly, as indicated by the growing number of green arrowheads in Fig. 2c–f. Along the fault plane (100), contrast of the white spots corresponding to Mn columns becomes weak, possibly due to the Mn migration in the Li layer. It is thus suspected that the Mn migration is related to formation of the defect, as the migration happens in the core of the defect.

A comparison of an experimental voltage-composition plot of a $Li/Li_2MnO_3$ cell (black line) with the calculated plot of a cell (colored lines) with a $Li_2MnO_3$ electrode containing defects is provided in Fig. 2g. The $b/6[110]$ defects are believed to be associated predominantly with the first step (4.89 V), that is, without oxygen loss, whereas the $c/2[001]$ defects are believed to be associated predominantly with the second step (5.03 V), that is, with oxygen loss. Without the $c/2[001]$ defects, the second step was calculated to occur at a slightly higher voltage (5.13 V). Figure 2h illustrates a schematic evolution of the $c/2[001]$ defects and a generalized mechanism by which oxygen species can be transported through a highly defective and faulted $Li_2MnO_3$ electrode structure before being released as fully oxidized $O_2$ gas at the surface.

Although dislocations that move transversely to a glide plane in metal-oxide structures are unusual, the $c/2[001]$ dislocation formed dynamically during electrochemical delithiation of $Li_2MnO_3$ can glide and climb with apparent ease. The evolution of these defects (indicated by the green bars) as delithiation progresses is highlighted in Fig. 3a–d; corresponding Fourier filtered images, showing only the (001) lattice planes to emphasize the perpendicular movement of the defects relative to the (001) planes, are shown in Fig. 3e–h. The dislocations glide progressively toward the right surface, as indicated by the changing position of the blue "half-cross" markers with increasing delithiation (reaction time). During this process, the distance between defects narrows until they merge and become one (Fig. 3d, h). The Li compositional gradient between the surface and the core (and thus the strain caused) could be the driving force for the climbing and gliding of the defects.

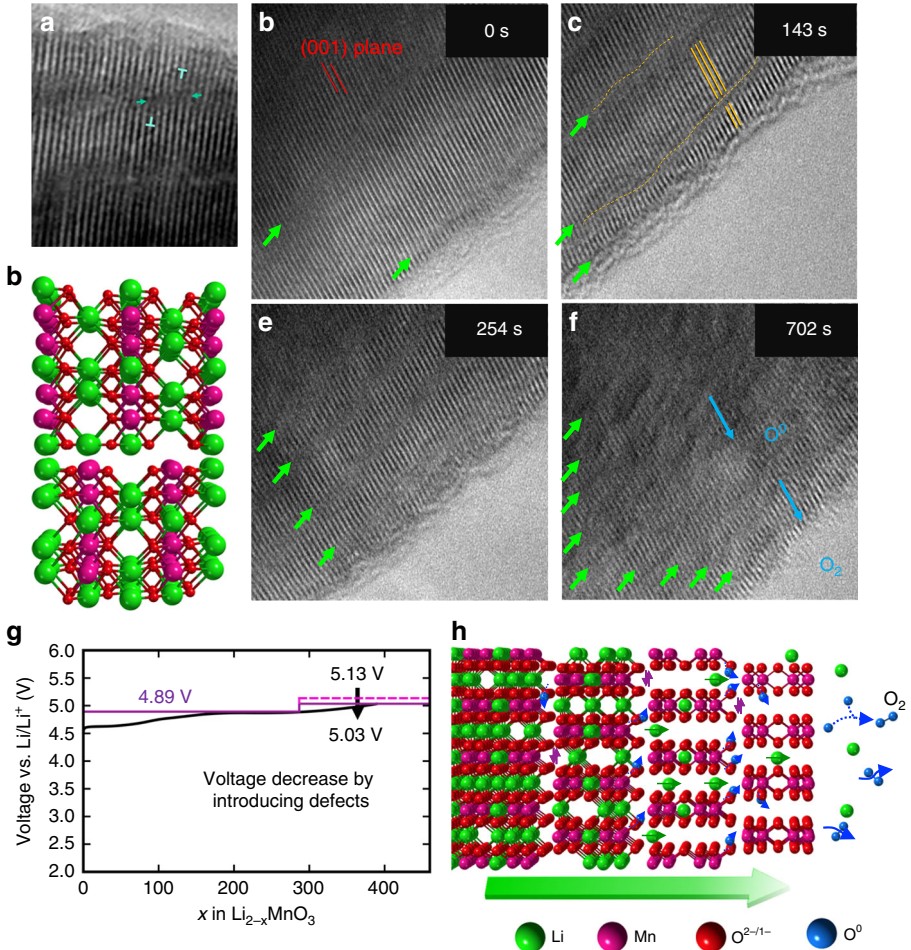

**Fig. 2** The form and motion of the second kind of defects upon delithiation. **a** Dissociated dislocation in $Li_2MnO_3$ with Burgers vector $c/2[001]$ formed dynamically in delithiation. **b** Atomic model of the dislocation containing lithium vacancies in both Li- and Li-Mn layers. **c–f** Defect density (green arrowheads) of the $Li_2MnO_3$ electrode increases as the delithiation process progresses. **g** Comparison of experimental voltage–composition plot of a $Li/Li_2MnO_3$ cell (black) with the calculated plot of cells containing $Li_2MnO_3$ electrodes with $b/6[110]$ defects dominating the first step (4.89 V) and $c/2[001]$ defects dominating the second step (5.03 V). Without the $c/2[001]$ defects, the second step would occur at voltage of 5.13 V. **h** Illustration of the $c/2[001]$ defects and a proposed mechanism by which oxygen is transported in a highly defective $Li_2MnO_3$ electrode structure and released at the surface

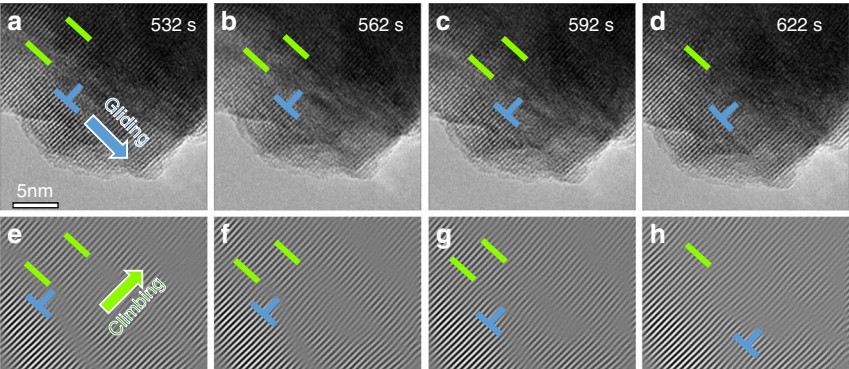

**Fig. 3** Dynamic gliding and transverse movement ("climbing") of a $c/2[001]$ dislocation during delithiation. **a–d** Time-lapsed high-resolution episcopic microscopy (HREM) images. **e–h** Corresponding Fourier-filtered images of **a–d**, showing only the (001) lattice plane fringes. Gliding, climbing, and merging of the dislocations towards the surface prompts the formation and release of $O_2$ gas. The scale bar is 5 nm

The creation of these defects during electrochemical delithiation are dependent on lithium-ion vacancies and, in the absence of oxidized manganese above $4^+$, on oxidized $O^{2-}$ species, as proposed in Fig. 2b and referred to as $O^0$ for simplicity and convenience. The extent to which the $O^{2-}$ species is oxidized (or hybridized) is not yet known. Furthermore, because the $c/2[001]$ dislocations are mobile and migrate towards the surface upon further delithiation, "trapped" oxidized $O^{2-}$ species in these structural defects can be transported from the interior of a $Li_2MnO_3$ crystal to the surface, where they can either be

released as $O_2$ gas or, alternatively in the case of a conventional lithium-ion cell, react with an organic liquid electrolyte solvent. With this information, we tentatively propose that a dislocation-assisted electrochemical reaction in which lithium is extracted from $Li_2MnO_3$ ultimately with oxygen release could proceed as follows:

(i)   the formation of $c/2[001]$-induced defects during lithium extraction,
(ii)  the formation of oxygen vacancies within the defects with oxidized $O^{2-}$ species residing at defect boundaries,
(iii) the gliding and "climbing" of the defects that transport the oxidized $O^{2-}$ species to the crystal surface and,
(iv)  combination reactions between $O^0$ species and the release of $O_2$ molecules at the surface.

A computerized schematic of this process is provided in Fig. 2h. Readers are encouraged to watch the videos of the in situ TEM experiments provided in the Supplementary Information section, in which the evolution and dynamic behavior of defects during electrochemical delithiation of $Li_2MnO_3$ electrodes can be observed in real time.

**Oxygen release confirmed by DFT calculations.** The release of oxygen from $Li_2MnO_3$ was also assessed by calculating the $O^0$ vacancy formation energy as a function of Li removal—a lower $\Delta E_{Vac}^{Form}$ value implying a more facile O extraction process, while a negative $\Delta E_{Vac}^{Form}$ value implies a spontaneous release of oxygen. The calculations, mapped graphically in Fig. 4a, show that the calculated oxygen-vacancy formation energy decreases with increasing Li removal, but remains largely positive over the compositional range $(0.0 < x < 1.0)$, suggesting that lithium extraction would have to be charge compensated by a partial oxidation of the oxygen ions without any oxygen release. Spontaneous oxygen release is predicted to occur only after a large amount of lithium has been extracted from an ideal $Li_2MnO_3$ structure, that is, $Li_{2-x}MnO_3$, $x > 1.5$. Such structural stability seems highly unlikely particularly in a practical lithium cell environment in which the highly oxidizing $Li_{2-x}MnO_3$ electrode would be in direct contact with a reactive electrolyte solvent. Figure 4b shows that the energy difference of $Li_{2-x}MnO_3$ with and without $c/2[001]$ defects becomes negative once approximately

one-half of the lithium ions have been extracted from $Li_2MnO_3$, thereby providing clues about the composition at which energetically favorable defects would form in an inert environment. The formation of the $c/2[001]$ defects will promote the $O_2$ release at a earlier stage. Of particular significance, however, is that the calculations indicate that the $c/2[001]$ defect boundary and the lithium-depleted structure (e.g., $x=1.875$) contains short O-O bonding distances (~1.5 Å) (Fig. 4c) relative to the non-bonding distance of ~3.1 Å in pristine $Li_2MnO_3$, consistent with earlier calculations reported by Benedek et al.[51], thereby giving credence to the mechanism suggested in this study.

## Discussion

In summary, two types of stacking faults and corresponding partial dislocations, formed during the electrochemical delithiation of $Li_2MnO_3$ electrodes have been identified by in situ TEM studies supported by DFT calculations. Defects with fault vector of $b/6[110]$ appear to have low activation energy and may contribute to reversible oxygen redox behavior. On the other hand, dissociated dislocations with Burgers vector of $c/2[001]$ are created at higher voltage (>4.5 V) and assist the transport of oxidized oxygen species to the electrode surface where $O_2$ is formed and released irreversibly. The study reveals an important connection between crystalline defects and the electrochemical behavior of lithium-rich metal-oxide materials, which may pave the way for further understanding and control of oxygen redox reactions, particularly in high capacity $Li_2MnO_3$-stabilized electrodes for lithium-ion batteries.

## Methods

**Synthesis of nanostructured $Li_2MnO_3$**. All the chemicals used in the work are analytically pure grade. Stoichiometric amounts of $Li_2CO_3$ and $MnCO_3$ precursor powders were thoroughly mixed and fired at 800 °C for 12 h. The heating rate was 2 °C/min and cooling rate was not controlled (furnace cooling). The obtained powder sample was ground and sieved for the subsequent characterization and electrochemical measurements.

**In situ TEM**. The open half-cell was constructed in an in situ electrical probing TEM holder (Nanofactory Instrument). This holder has a dual-probe design, that is, one Au rod is used as the sample holder with a small amount of nanostructured $Li_2MnO_3$ attached to its tip; on the other side, a STM tungsten (W) probe driven by Piezo-motor capable of 3D positioning with a step size of 1 nm was used to mount Li metal. The W probe tip was scratched by Li metal strip and then affixed on the

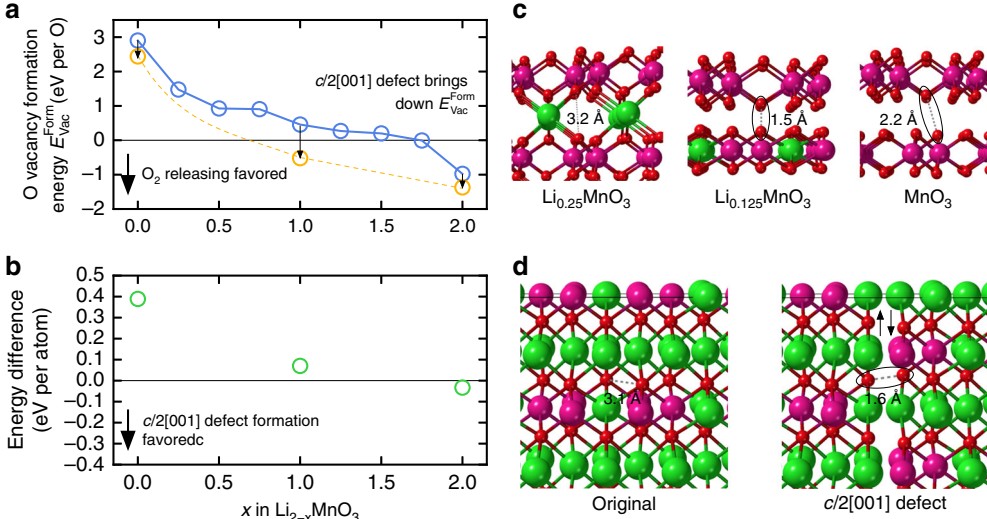

**Fig. 4** The impact of defect formation on oxygen loss during delithiation of $Li_2MnO_3$. **a** Calculated O vacancy formation energy as a function of Li removal before and after $c/2[001]$ defects are introduced to the system. **b** Energy difference between $Li_2MnO_3$ structures with and without $c/2[001]$ defects. **c** Oxygen–oxygen interactions in $Li_{2-x}MnO_3$ systems without defects. **d** Oxygen–oxygen interactions in systems with $c/2[001]$ defects

TEM holder inside an Ar-filled glove box. With an airtight cover, the TEM holder was transferred to TEM column with limited exposure to air (~10 s), where a layer of lithium oxide was grown on the surface of Li metal and acted as a solid electrolyte for the nano-cell lithium-ion batteries. When the Au rod was positively biased to 5 V, discharging for nanostructured $Li_2MnO_3$ nanoparticles occurred, corresponding to the electrochemical delithiation. The in situ TEM is performed on a field-emission JEOL-2100F transmission electron microscope, operated at 200 kV. The images are collected by a Gatan GIF Camera. The drift of the collected images is corrected mathematically by the IMOD software.

**First-principle calculations**. DFT calculations reported in this study were conducted via the Vienna Ab-initio Simulation Package with the projector augmented wave potentials and the Perdew–Becke–Ernzerhof approximation was employed to the exchange-correlation potential. A plane wave basis with a cutoff energy of 520 eV and $\Gamma$-centered $k$-meshes with a density of 8000 $k$-points per reciprocal atom were used for all calculations. All calculations were spin polarized, with Mn atoms initialized in a high-spin configuration and relaxed to self-consistency.

## Data availability
The authors declare that all data supporting the findings of this study are available within the paper and its Supplementary Information files.

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

## Acknowledgements

Q.L., Z.Y, Y.X., E.L., M.M.T., C.W., and V.P.D. were supported as part of the Center for Electrochemical Energy Science, an Energy Frontier Research Center funded by the US Department of Energy (DOE), Office of Science, Basic Energy Sciences under Award # DE-AC02-06CH11357. J.W. and V.P.D. were also supported by the Samsung Advanced Institute of Technology (SAIT)'s Global Research Outreach (GRO) Program. This work made use of the EPIC facility of Northwestern University's NUANCE Center, which has received support from the Soft and Hybrid Nanotechnology Experimental (SHyNE) Resource (NSF ECCS-1542205); the MRSEC program (NSF DMR-1720139) at the Materials Research Center; the International Institute for Nanotechnology (IIN); the Keck Foundation; and the state of Illinois, through the IIN. We acknowledge the computing resources from (i) the National Energy Research Scientific Computing Center, a DOE Office of Science User Facility supported by the Office of Science of the DOE under contract DE-AC02-05CH11231; and (ii) Blues, a high-performance computing cluster operated by the Laboratory Computing Resource Center at Argonne National Laboratory. Q.L. gratefully acknowledges the supporting of National Natural Science Foundation of China (Grant No. 51702207). J.W. was supported by the Fundamental Research Funds for the Central Universities (WUT: 2019III012GX).

## Author contributions

Q.L., Z.Y., and J.W. conceived the project. Q.L., Y.X., J.W., and V.P.D. performed the in situ TEM and interpretation). Z.Y. and C.W. conducted DFT simulations. E.L. and M.M.T. conducted materials synthesis, battery measurements, and data interpretation. All of the authors contributed to the writing to the manuscript before submission.

## Additional information

**Competing interests:** The authors declare no competing interests.

