## [Peer Review File · Nature Communications]

Reviewers' comments:

Reviewer #1 (Remarks to the Author):

In the work, two types of defects have been characterized and studied by in-situ TEM. The study of dynamically formed defects in delithiation provides novel insights on the role of defects. Meanwhile, anionic redox, especially the oxygen redox is an inspiring concept of the cathode materials for high energy Li ion battery. As one of the original system featured with oxygen redox, detailed oxygen oxidation/extraction reaction process of Li_2MnO_3 upon high voltage (related to activation) is still unclear and draws continuous attention. However, the lacking of proper real time atomic scale observation techniques has been hindering the clear determination of the process, until now the in-situ TEM fills the gap. In this manuscript, the authors explore the mechanism from a crystallographic defect perspective and demonstrated that both the oxygen removal and transport have been facilitated by a planar defect. A combination of in-situ TEM and DFT study was conducted and the manuscript is well written. I would acknowledge the progress achieved by this study and recommend it for publication if the authors can clarify the following points:

1. $b/6[110]$ stacking fault has commonly been observed in Li_2MnO_3 crystals (especially processed at a relatively low temperature). It is required to give a clear explanation of the difference/similarity between the dynamically formed $b/6[110]$ stacking fault.

2. Oxygen vacancy has been examined in the previous studies (e.g.

DOI: 10.1002/aenm.201400498)

with predicted oxygen release point at $x=1$ of $\text{Li}_{2-x}\text{MnO}_3$, contrary to the DFT results here in this study: $x=1.5$. The authors need to show more proofs to validate the delay of O release.

3. The authors are asked to show more evidence on the redox evolution of Mn and O (besides the oxygen release) during the delithiation process to give the audience a sense of the reaction whole picture.

Minor revisions:

Page 8,

line 170-171: "ABC1 in pristine Li_2MnO_3 to an intermediate AC2C1 arrangement and subsequently to AC2B". The labels and captions in Fig.1 should match to this.

cite more relevant papers such as: Chemistry of Materials 25 (11), 2319-2326; Nano letters 12 (10), 5186-5191, etc

Reviewer #2 (Remarks to the Author):

The manuscript titled "Dynamic imaging of crystalline defects in Li_2MnO_3 electrodes during electrochemical activation to high voltage " investigates crystalline defects generated in lithium-metal-oxide electrodes during cycling of lithium-ion batteries. Particularly, authors monitored dynamically by TEM during delithiation two types faults in Li_2MnO_3 - one with a fault vector $b/6[110]$ and low mobility; and another planar defect of with fault vector $c/2[001]$ exhibiting high gliding and transverse mobility. Authors speculated the second type of defects were associated with oxygen loss. These observations are potentially important for understanding capacity loss during the initial charge/discharge in high capacity lithium metal oxide electrodes, and are of interest to scientific community. However, it is premature to draw a definite conclusion based on very limited observations and analysis.

I recommend major revision with additional experiments before it can be further considered for publication. The followings are the major concerns that need to be addressed in the revision:

(1) Authors have used various terms to describe the second type of defects, "dynamic defects", "planar defects with fault vector $c/2 [001]$ ", "vector $c/2 [001]$ type faults", "partial dislocation with the burgers vector $c/2 [001]$ ". Despite those efforts, it is not clear what are nature of the defect! Particularly, what is the fault plane? atomic structure? how is the "climbing" of the defects accomplished? Since the partial dislocation is associated with the planar fault, does the "climbing" lead to a change of the fault plane?

(2) The $c/2 [001]$ type defects appear always in pair with the fault vectors of the opposite sign. (This observation was verified by careful inspection of images in Fig.2d and 2e, as well as accompanying videos). So, introducing the defects during delithiation does not lead to the reduction or increase of the lattice (001) planes, the proposed models in Fig.2h, therefore, is inconsistent with the experimental observation.

(3) Even if the proposed model was correct, it is not clear how big impact would be to the overall oxygen loss during the delithiation, given that the side surface of a TEM sample is very small in comparison to the top/bottom surface of the TEM sample. One would expect the major loss mechanism would be related to oxidation of oxygen ions and release of oxygen gas through the top/bottom surface. An estimation of the gas loss efficiency by the proposed mechanism would be useful.

(4) It is well known that the Li_2MnO_3 is very sensitive to high electron beam irradiation. The impact of the electron beam irradiation is not clear to the observation reported. Some "dynamic defects" were seen after few minutes of the beam exposures at a high magnification, the impact of such effect (e.g., loss of Li by electron beam) cannot be ignored. A controlled experiment is needed to verify the findings.

Reviewer #3 (Remarks to the Author):

The authors present a superb and difficult study of the operando release of oxygen during delithiation of Li_2MnO_3 , the model compound, basis of the Li-Rich family. They combined TEM imaging and DFT calculations to propose a mechanism in order to explain the voltage plateau and the global shape of the electrochemical curve. They identified defects, behaving differently when Li is removed. The communication is written in such a way that the reader can assess properly the conclusions and all figures are perfectly clear, convincing (as well as associated movies) so that the global quality of the paper is extremely high. The implications sustained by the discovered mechanism are so important that a publication in Nature Comm. would profit largely to the community.

There are however modifications and precisions that need to be given so that it can be the case. First, a few points on the mechanism:

- It is important for the community to reconcile (if possible) the bulk/surface discrepancy as still noticed by some authors (Teufl et al. JES 165 A2718 (2018)) with the mechanism exposed by the authors. It is in particular unclear if the phase at the surface following the last step (Figure 2h) is the spinel phase? In order to be complete with the recent literature, is the proposed mechanism in accordance with the O₂-Li-rich material discovered by Zuo et al (Advanced Materials, 30, 1707255 (2018)), that does not seem to be generating O₂ release. Is $c/2$ climbing prevented in such system?

- What is the relation between "gliding" of the $c/2$ and the oxygen release by the surface? It is unclear why the authors disregarded totally the influence of the (001) surface as a O₂ release area.

- Still on this climbing, Fig3a-d are indicted as climbing in the opposite direction to the surface (at

the bottom of the images). Is it compatible with the diffusion of oxygen species to the surface (L199-202)?

- Although clearly very challenging, chemical information is not provided (Oxygen, Mn or Li). Do authors have such experimental proof? In particular, the Li content matters a lot in the initiation of the C/2 dislocation and maybe it could be accessed through EELS at the Oxygen K edge?
- Nothing is said about the Mn migration in the Li layer. This is an important phenomenon and the authors should comment about it and its relation to the dislocations.
- L150-151(suppl): the values are barely negative for the formation energy of the c/2 dislocation. In fact, the figure shows positive values till $x=1.5$. In L168, the first step is the formation of dislocations. But these are not energetically favourable till a large depletion of lithium. So how can it be the first step? Would it be possible that the depletion is higher inside the crystals (core-shell Li extraction), hence making the creation of c/2 dislocation possible (before the electrochemical $x=1.5$ composition)?
- The dynamical behaviour of defects is quite well observed by the authors but it is not clear what the driving force for this behaviour is? Why do dislocations move to the surface? Is it due to Li compositional gradient?

Other more technical questions but still important to the paper:

- One cannot observe the Li₂O solid electrolyte on the TEM images. Is it present at the surface? Was the crystal in contact somewhere else on its surface? If yes, how did it influence the ion migration, hence the diffusion of oxygen species (any orientation dependence)?
- The TEM experimental details need to be developed further (in supplementary), considering most important results are obtained with this technique. The microscope should be mentioned (image corrected or not?) as well as the high tension (essential!). Since image quality is also important, the camera used in the study should be indicated (any special mode, drift correction, etc...)?
- It is a well known consequence of electron beams that it can produce defects. It can even, because of beam damages following oxygen release, produce phase changes, such as new crystal symmetries. The authors should develop the verifications they did so that the electron beam was not involved in the dynamical changes observed.
- It would also be interesting to get more info on the electrochemical response during the operando experiment: if direct biasing to 5V is applied, what is the measured current? How does it influence the results since most of the case cycling is operated in galvanostatic mode? What is the equivalent C rate? Discussing implication on the charging process and O₂ release would be useful to the reader to assess the extent of the discovery.
- TEM image contrast is particularly difficult to interpret. It is usually admitted that in order to retrieve information with confidence, simulated images should always be produced. They could in particular allow the reader to be more confident with the interpretation of the authors about the loss of contrast at the dislocation. In particular, the width of the stacking fault could be compared with the proposed model.
- What is the fundamental reason for atomic resolution not to be achieved in in situ images of the c/2 dislocation ((Figure 2) whereas it is the case for a (001) stacking fault (b/6, Figure S5)?

-

A few more miscellaneous points:

- L22: "in operando" → "operando" the latin ablative already includes "in"
- L85: in-situ → in situ
- L107 (Suppl). : not clear what "Enum" is.
- L113 (suppl): Fig S4 is not linked to "energy configuration" but d spacings. Please explain or modify.
- L114 (suppl): Fig 2 does not show explicitly stacking sequences (maybe Fig 1 ?).
- L123(suppl): I do not see the majority ABC stacking sequence in Fig. 2.
- L125 (suppl): clearly should be Fig2g instead Fig3g.
- L132(suppl): there is no Fig5a...probably 4a...
- L138(suppl): "then" should be "than"
- L150(suppl): Fig5b → Fig4b

- Please check all Figure references in this part of the Supplementary.
- L168(suppl): Fig2h

Materials Science and Engineering

2220 Campus Drive, Cook Hall 1154

Evanston, Illinois 60208-3108

(847) 491-7807

Jinsong-wu@northwestern.edu

NORTHWESTERN
UNIVERSITY

December 6, 2018

Reviewer 1:

In the work, two types of defects have been characterized and studied by in-situ TEM. The study of dynamically formed defects in delithiation provides novel insights on the role of defects. Meanwhile, anionic redox, especially the oxygen redox is an inspiring concept of the cathode materials for high energy Li ion battery. As one of the original system featured with oxygen redox, detailed oxygen oxidation/extraction reaction process of Li_2MnO_3 upon high voltage (related to activation) is still unclear and draws continuous attention. However, the lacking of proper real time atomic scale observation techniques has been hindering the clear determination of the process, until now the in-situ TEM fills the gap. In this manuscript, the authors explore the mechanism from a crystallographic defect perspective and demonstrated that both the oxygen removal and transport have been facilitated by a planar defect. A combination of in-situ TEM and DFT study was conducted and the manuscript is well written. I would acknowledge the progress achieved by this study and recommend it for publication if the authors can clarify the following points:

Authors Response: We appreciate the reviewer's high recognition of our contribution.

1. $b/6[110]$ stacking fault has commonly been observed in Li_2MnO_3 crystals (especially processed at a relatively low temperature). It is required to give a clear explanation of the difference/similarity between the dynamically formed $b/6[110]$ stacking fault.

Authors Response: We agree with the reviewer that the $b/6[110]$ stacking fault has been previously observed and reported. All the previous studies have been focused on the study of the static $b/6[110]$ stacking faults in intact or fully charged (delithiated) Li_2MnO_3 . In this study, we studied the $b/6[110]$ stacking faults dynamically formed during the delithiation and disclosed its relationship with Li extraction. We observed dynamic change of the lattice plane stacking (to form the stacking fault) driven by the $b/6[110]$ partial dislocation.

2. Oxygen vacancy has been examined in the previous studies (e.g. DOI:10.1002/aenm.201400498) with predicted oxygen release point at $x=1$ of $\text{Li}_{2-x}\text{MnO}_3$, contrary to the DFT results here in this study: $x=1.5$. The authors need to show more proofs to validate the delay of O release.

Authors Response: Negative oxygen vacancy formation energy, which means the spontaneous formation of oxygen vacancy in the structure, has been widely used as the sign of the initiation of oxygen release. Previous studies of oxygen vacancy formation as a function of Li content have typically used a fixed stacking sequence (i.e., that of Li_2MnO_3), and examined the interdependence of oxygen release and delithiation. In contrast, we consider various stacking sequences (not only that of the original Li_2MnO_3 phase), and find that for delithiation in the range of approximately $1 < x < 1.5$, the system shows an energetic preference for these alternative stacking sequences. Since the energies of the delithiated $\text{Li}_{2-x}\text{MnO}_3$ phases in our paper are lower than those of previous studies, the resulting oxygen vacancy formation energies are higher, and hence the spontaneous release of oxygen is delayed to higher values of x . These DFT results are confirmed by our TEM observations of stacking faults for these highly delithiated compositions.

3. The authors are asked to show more evidence on the redox evolution of Mn and O (besides the oxygen release) during the delithiation process to give the audience a sense of the reaction whole picture.

Authors Response: We monitored the Mn/O valence evolution through the whole Li_2MnO_3 delithiation by checking the magnetic moments of Mn and O as shown in Fig. R1. This method of examining redox evolution has been successfully applied to compounds such as $\text{Li}_{1.25}\text{Mn}_{0.5}\text{Nb}_{0.25}\text{O}_2$ (1), Li_5FeO_4 (2), and $\text{Li}_4\text{Mn}_2\text{O}_5$ (3). We can see a clear trend that Mn ions generally stay with their 4+ charge state (i.e., the Mn magnetic moments do not significantly change from the value in the completely lithiated Li_2MnO_3 compound), while more and more O^{2-} ions have been oxidized to O^{1-} (i.e., the magnetic moments of oxygen significantly rise for delithiated compositions, compared to the $m=0$ value in the Li_2MnO_3 compound). We also see from our oxygen vacancy energy calculations that a fraction of oxygen ions will be further oxidized to O^0 and leave the structure as gas phase O_2 .

(1) Seo, D.-H.; Lee, J.; Urban, A.; Malik, R.; Kang, S.; Ceder, G. Nat. Chem. 2016, 8 (7), 692–697.

(2) Zhan, C.; Yao, Z.; Lu, J.; Ma, L.; Maroni, V. A.; Li, L.; Lee, E.; Alp, E. E.; Wu, T.; Wen, J.; et al. Nat. Energy 2017, 2 (12), 963–971.

(3) Yao, Z.; Kim, S.; He, J.; Hegde, V. I.; Wolverton, C. Sci. Adv. 2018, 4 (5), eao6754

Figure R1. Mn and O valence state evolution during the delithiation of Li_2MnO_3 .

Minor revisions:

Page 8,

line 170-171: “ABC1 in pristine Li_2MnO_3 to an intermediate AC2C1 arrangement and subsequently to AC2B”. The labels and captions in Fig.1 should match to this.

Authors Response: Thanks. The labels in Fig.1 has been updated.

cite more relevant papers such as: Chemistry of Materials 25 (11), 2319-2326; Nano letters 12 (10), 5186-5191, etc

Authors Response: Thanks. We have added the references into the revised manuscript.

Reviewer #2 (Remarks to the Author):

The manuscript titled "Dynamic imaging of crystalline defects in Li₂MnO₃ electrodes during electrochemical activation to high voltage" investigates crystalline defects generated in lithium-metal-oxide electrodes during cycling of lithium-ion batteries. Particularly, authors monitored dynamically by TEM during delithiation two types faults in Li₂MnO₃ - one with a fault vector $b/6[110]$ and low mobility; and another planar defect of with fault vector $c/2[001]$ exhibiting high gliding and transverse mobility. Authors speculated the second type of defects were associated with oxygen loss. These observations are potentially important for understanding capacity loss during the initial charge/discharge in high capacity lithium metal oxide electrodes, and are of interest to scientific community. However, it is premature to draw a definite conclusion based on very limited observations and analysis.

I recommend major revision with additional experiments before it can be further considered for publication. The followings are the major concerns that need to be addressed in the revision:

Authors Response: We appreciate the reviewer's positive comments.

(1) Authors have used various terms to describe the second type of defects, "dynamic defects", "planar defects with fault vector $c/2[001]$ ", "vector $c/2[001]$ type faults", "partial dislocation with the burgers vector $c/2[001]$ ". Despite those efforts, it is not clear what are nature of the defect! Particularly, what is the fault plane? atomic structure? how is the "climbing" of the defects accomplished? Since the partial dislocation is associated with the planar fault, does the "climbing" lead to a change of the fault plane?

Authors Response: We agree that our original descriptions were unnecessarily confusing, and have modified/clarified the manuscript. The defect is a planar defect bounded by a partial dislocation; specifically, the planar defect can be described as an antiphase boundary (while the burgers vector of the partial dislocation is $1/2[001]$, so the fault vector of the antiphase boundary is also $1/2[001]$). The 'fault plane' of the antiphase boundary is in the (100) plane. The atomic structure of the defect is shown in Fig.2b. Climbing refers to the movement of the defect across the (100) plane, while gliding refers to the movement in the (100) plane. Following the kind suggestions of the reviewer, we have corrected all of the related parts and given a clear description of the defect (defined as a *dissociated dislocation with Burgers vector of $c/2[001]$*), by adding to the revised manuscript "More precisely speaking, the defect is a dissociated dislocation consisting of an antiphase boundary (with fault vector of $1/2[001]$) and the partial dislocation bounded to the antiphase boundary. The 'fault plane' of the antiphase boundary is in the (100) plane with the atomic structure of the defect shown in Fig.2b. Climbing of the dislocation refers to the movement of the defect across the (100) plane, while gliding refers to the movement in the (100) plane".

(2) The $c/2$ [001] type defects appear always in pair with the fault vectors of the opposite sign. (This observation was verified by careful inspection of images in Fig.2d and 2e, as well as accompanying videos). So, introducing the defects during delithiation does not lead to the reduction or increase of the lattice (001) planes, the proposed models in Fig.2h, therefore, is inconsistent with the experimental observation.

Authors Response: The images and video show only Co-columns (while the Li-columns has no contrast in the current HREM observation). In experiment, it shows that Co-columns (and thus the (001) lattice plane) do not reduce or increase (but only shift to form an antiphase boundary). In the proposed model (Fig.2H), the Co-columns shift to form the antiphase boundary (when Li-ions layers are extracted), which is consistent to the experimental results.

(3) Even if the proposed model was correct, it is not clear how big impact would be to the overall oxygen loss during the delithiation, given that the side surface of a TEM sample is very small in comparison to the top/bottom surface of the TEM sample. One would expect the major loss mechanism would be related to oxidation of oxygen ions and release of oxygen gas through the top/bottom surface. An estimation of the gas loss efficiency by the proposed mechanism would be useful.

Authors Response: Thanks for your helpful comments. The gliding and climbing of the partial dislocation, which happen in a speedy manner, will transport the reduced oxygen toward all of the surfaces, including the top/bottom surfaces (although in the TEM observation, we can merely observe those dislocations moving toward the side surface). The defect will thus prompt the extraction and transport of O toward all the surfaces, formation O-O bonding and release of O₂.

(4) It is well known that the Li₂MnO₃ is very sensitive to high electron beam irradiation. The impact of the electron beam irradiation is not clear to the observation reported. Some “dynamic defects” were seen after few minutes of the beam exposures at a high magnification, the impact of such effect (e.g., loss of Li by electron beam) cannot be ignored. A controlled experiment is needed to verify the findings.

Authors Response: Thanks. We have performed the suggested controlled experiment to observe carefully the influence of the electron irradiation. The influence of the electron irradiation to the defects generation and movement (as observed in delithiation) is small, as shown in Video R1. The video shows Li₂MnO₃ under electron irritation for more than 30 minutes and one cannot observe the defects as observed in delithiation.

Reviewer #3 (Remarks to the Author):

The authors present a superb and difficult study of the operando release of oxygen during delithiation of Li_2MnO_3 , the model compound, basis of the Li-Rich family. They combined TEM imaging and DFT calculations to propose a mechanism in order to explain the voltage plateau and the global shape of the electrochemical curve. They identified defects, behaving differently when Li is removed. The communication is written in such a way that the reader can assess properly the conclusions and all figures are perfectly clear, convincing (as well as associated movies) so that the global quality of the paper is extremely high. The implications sustained by the discovered mechanism are so important that a publication in Nature Comm. would profit largely to the community.

Authors Response: We really appreciate the reviewer's high recognition of our contribution.

1. There are however modifications and precisions that need to be given so that it can be the case.

First, a few points on the mechanism:

- It is important for the community to reconcile (if possible) the bulk/surface discrepancy as still noticed by some authors (Teufl et al. JES 165 A2718 (2018)) with the mechanism exposed by the authors. It is in particular unclear if the phase at the surface following the last step (Figure 2h) is the spinel phase? In order to be complete with the recent literature, is the proposed mechanism in accordance with the O₂-Li-rich material discovered by Zuo et al (Advanced Materials, 30, 1707255 (2018)), that does not seem to be generating O₂ release. Is c/2 climbing prevented in such system?

Authors Response: Thanks. The formation of spinel phase on surface is possible, although it is hard to have a direct and clear evidence in the current in-situ TEM observations/images. The images collected by the in-situ sample holder were taken along a high order zone axis, which was not straightforward to analysis.

In the report by Zuo et al.³ (Ref. Advanced Materials, 30, 1707255 (2018)) O₂-type LiMO_2 composite with a single-layer O₂- Li_2MnO_3 superstructure, the transformation from the layer to the spinel phase is unfavorable because the oxygen lattice rearrangement requires all the Mn-O bonds in the O₂-type structure to break. (Ref. doi: 10.1149/1.1392514; J. Electrochem. Soc. 1999 volume 146, issue 10, 3560-3565), thus the O₂- Li_2MnO_3 superstructure could prevent the formation of oxygen-oxygen dimers, possibly will also make the c/2 climbing difficult.

2. What is the relation between "gliding" of the c/2 and the oxygen release by the surface? It is unclear why the authors disregarded totally the influence of the (001) surface as a O₂ release area.

Authors Response: The gliding and climbing of the partial dislocation will transport the reduced oxygen toward all of the surfaces (gliding will transport the activated oxygen to the (001) surfaces, while the climbing will transport it to the (100) and/or (010) surface).

3. Still on this climbing, Fig3a-d are indicted as climbing in the opposite direction to the surface (at the bottom of the images). Is it compatible with the diffusion of oxygen species to the surface (L199-202)?

Author's response: Climbing (and gliding) of the partial dislocation occurs in the direction determined by strain possibly formed by lithium-ion gradient (in delithiation). According to the local strain, climbing can occur in various direction. However, only those moving toward the surface contribute to the release of O₂ (and thus the formation of possible spinel phase on the surface).

4. Although clearly very challenging, chemical information is not provided (Oxygen, Mn or Li). Do authors have such experimental proof? In particular, the Li content matters a lot in the initiation of the C/2 dislocation and maybe it could be accessed through EELS at the Oxygen K edge?

Author's response: Thanks for the suggestions. In a delithiated sample (data was collected right after the in-situ experiment), the elemental distribution of Li, O and Mn from the surface to the inner part has been identified by EELS, as shown in Figure R2. The intensity of lithium K-edge gradually increases from the vacuum to the inner part of the nanoparticle along the green line. The intensity of O K-edge gradually decreases from the surface to the inside region (Fig. R1(f)), indicating that the oxygen content in the area closed to the surface is higher than that of inner part. Meanwhile, the intensity of Mn L-edge has almost no changes from the surface to the inner area of the crystal. This supports the model proposed by the work on the O₂ release.

Figure R2. (a) Typical TEM image of the delithiated region. (b) High-resolution of the delithiated region squared by the red color in (a). (c) The Li distribution along the green line were shown in (e). (d) the O and Mn distribution along the green line were shown in (f).

5. Nothing is said about the Mn migration in the Li layer. This is an important phenomenon and the authors should comment about it and its relation to the dislocations.

Author's response: Unlike the defects, Mn migration (occurred normally in an un-ordered manner) requires significantly higher resolution to resolve. Although we did not observe such migration directly, from the contrast change in along the “fault plane” (Fig.2), there is possible sign of Mn migration (in the image, the white spots correspond to Mn columns; such white spots have a much lower and diffused contrast in the ‘fault plane’ area). However, as the proof is not very clear, we did not discuss this in the manuscript. In the revised manuscript, we have added “Along the fault plane (100), contrast of the white spots corresponding to Mn columns becomes weak, possibly due to the Mn migration in the Li layer. It is thus suspected that the Mn migration is related to formation of the defect, as the migration happens in the core of the defect”.

6. L150-151(suppl): the values are barely negative for the formation energy of the $c/2$ dislocation. In fact,

the figure shows positive values till $x=1.5$. In L168, the first step is the formation of dislocations. But these are not energetically favourable till a large depletion of lithium. So how can it be the first step? Would it be possible that the depletion is higher inside the crystals (core-shell Li extraction), hence making the creation of $c/2$ dislocation possible (before the electrochemical $x=1.5$ composition)?

Author's response: Thanks for the kind comment. We agree with the reviewer that, in reality, Li depletion is localized (e.g. areas on the particle surface) which can facilitate the formations of the $c/2$ dislocation to occur (dislocation itself is also a localized defect). In a certain localized area, the Li depletion may already reach (or above) $x=1$, thus facilitates the formation of the dislocation. We then made corresponding changes to the supporting information as follows: “As shown in Fig.4b, the energy difference decreases from 0.39 eV/atom ($x = 0$) to 0.06 eV/atom ($x = 1$), and becomes negative at $x = 2$. In reality, Li depletion is localized (e.g. areas on the particle surface) which can facilitate the formations of the $c/2$ dislocation.”

7. The dynamical behaviour of defects is quite well observed by the authors but it is not clear what the driving force for this behaviour is? Why do dislocations move to the surface? Is it due to Li compositional gradient?

Author Response: Thanks! We agree with the reviewer and the Li gradient difference between the surface and the core is one of the most important reasons for the defect gliding. The dislocations tends to move from the Li-ion rich area toward the li-ion poor area. We believe the driving force is the strain generated due to the Li compositional gradient. We have thus added to the discussion in the revised manuscript: “Li compositional gradient between the surface and the core (and thus the strain caused) could be the driving force for the climbing and gliding of the defects.”.

Other more technical questions but still important to the paper:

8. One cannot observe the Li_2O solid electrolyte on the TEM images. Is it present at the surface? Was the crystal in contact somewhere else on its surface? If yes, how did it influence the ion migration, hence the diffusion of oxygen species (any orientation dependence)?

Author Response: In our experiment, the contacting point of Li_2O (as solid electrolyte) and Li_2MnO_3 are relatively far away from the area used for in-situ TEM observation. This is because Li_2O is unstable under electron irradiation (thus we intentionally avoid the direct electron illumination on Li_2O). This is why Li_2O is not observed in the TEM images. The way of how Li_2O contact/touch Li_2MnO_3 may affect the Li compositional gradient near the particle/ Li_2O interface, yet the influence should be local and won't affect the overall Li/O-ion migration in the particle.

9. The TEM experimental details need to be developed further (in supplementary), considering most important results are obtain with this technique. The microscope should be mentioned (image corrected or

not?) as well as the high tension (essential!). Since image quality is also important, the camera used in the study should be indicated (any special mode, drift correction, etc...)?

Author Response: Those information asked by the reviewer are now included in the supplementary information: “The *in-situ* TEM is performed on a field-emission JEOL-2100F transmission electron microscope, operated at 200 kV. The images are collected by a Gatan GIF Camera. The drift of the collected images is corrected mathematically by IMOD software. ”

10. It is a well known consequence of electron beams that it can produce defects. It can even, because of beam damages following oxygen release, produce phase changes, such as new crystal symmetries. The authors should develop the verifications they did so that the electron beam was not involved in the dynamical changes observed.

Author's response: Thanks for your helpful comment. As in the reply to reviewer 2 (point 4), we have performed controlled experiment to exclude the impact of electron irradiation to the generated defects related to oxygen release, as shown in the Video R1.

11. It would also be interesting to get more info on the electrochemical response during the operando experiment: if direct biasing to 5V is applied, what is the measured current? How does it influence the results since most of the case cycling is operated in galvanostatic mode? What is the equivalent C rate? Discussing implication on the charging process and O₂ release would be useful to the reader to assess the extent of the discovery.

Author Response: In the nanobattery holder system (we used in the *in-situ* TEM), the current was in the range of tens to hundreds of nano-Ampere. However, due to the extra current generated by incident high energy electrons (from the TEM gun), it is hard to measure accurately the current. We appreciate the reviewer's kind suggestions and will take these into considerations in the future development of related *in-situ* TEM instruments.

12. TEM image contrast is particularly difficult to interpret. It is usually admitted that in order to retrieve information with confidence, simulated images should always be produced. They could in particular allow the reader to be more confident with the interpretation of the authors about the loss of contrast at the dislocation. In particular, the width of the stacking fault could be compared with the proposed model.

Author Response: We have performed HREM images simulation for perfect Li₂MnO₃ crystal and delithiated Li₂MnO₃ with defect. The images are simulated along the [100] zone axis as shown in Fig.R3. When the sample thickness is around 50 nm and defocus value is around 25 nm, the white dots in the

image correspond to Co-column. The shift of the Co-column in the delithiated Li_2MnO_3 leads to the

formation of the antiphase boundary.

(A)

(B)

Figure R3. Simulated HRTEM images of the perfect Li_2MnO_3 (A), and defected Li_2MnO_3 (B). The white spots in the simulated images as outlined by red frame correspond to Co-columns. The formation of ‘antiphase boundary’ like defects can be clearly seen (by comparing A and B).

13. What is the fundamental reason for atomic resolution not to be achieved in in situ imag

es of the $c/2$ dislocation ((Figure 2) whereas it is the case for a (001) stacking fault (b/6, Figure S5)?

Author Response: The fundamental reason is the crystalline orientation of Li_2MnO_3 being studied by in-situ TEM. Although it is not easy to find a crystal oriented along a low-order zone axis, good atomic resolution images can be observed when it is closed to the [100] zone axis (Figure S5).

A few more miscellaneous points:

- L22: “in operando” → “operando” the latin ablative already includes “in”

- L85: in-situ → in situ
- L107 (Suppl). : not clear what “Enum” is.

Ans: Enum is the name of an algorithm/code for enumerating/generating all derivative superstructures for arbitrary parent structures and for any number of atom types. We use this code to enumerate superlattices and atomic configurations in a geometry-independent way. We have added this definition into the revised manuscript.

- L113 (suppl): Fig S4 is not linked to “energy configuration” but d spacings. Please explain or modify.

Ans: This is a typo. “Fig S4” should be read as “Fig.S3”. It has been corrected.

- L114 (suppl): Fig 2 does not show explicitly stacking sequences (maybe Fig 1 ?).

Ans: Thanks. They are corrected.

- L123(suppl): I do not see the majority ABC stacking sequence in Fig. 2.
- L125 (suppl): clearly should be Fig2g instead Fig3g.
- L132(suppl): there is no Fig5a...probably 4a...
- L138(suppl): “then” should be “than”
- L150(suppl): Fig5b → Fig4b
- Please check all Figure references in this part of the Supplementary.
- L168(suppl): Fig2h

Author Response: Many thanks. All of the typos/corrections are corrected.

Reviewer #4 (Remarks to the Author):

This manuscript report detailed investigation on the oxygen loss from the Li_2MnO_3 during the electrochemical charge process using experimental and theoretical studies.

Since the initial introduction by the co-author in 1991 (Ref. 17), this Li-excess oxide has been extensively investigated as a promising cathode material with high charge capacity and high voltage. After the publication of Ref. 22 in 2007, the research on Li_2MnO_3 as a stable composite material with layer oxide LiMO_2 ($M = \text{Mn}, \text{Co}, \text{Ni}$), the research on this material has rapidly increased with the expectation of developing high capacity cathode material beyond LiMO_2 . Nevertheless, after 27 years of extensive academic research and commercial application attempts, this material is now viewed as not realistic cathode material candidate for commercial Li ion battery applications. Considering the extensive attempts and reports, the motivation of the manuscript is not well aligned with the battery community experience.

Nevertheless, the detailed underlying reasons for rapid degradation has not been fully understood after extensive research even though there is a reasonable consensus on the failure mechanisms of Li_2MnO_3 under electrochemical cycling conditions. From the perspective of fundamental materials science at atomic scale, the reported experimental finding and theoretical modeling results provide deeper insights on the underlying failure mechanisms. In a sense, these findings represent a helpful "tombstone analysis" of why this material has eventually failed to deliver the initial promise of high capacity cathode material candidate. Such analysis is a valuable finding, especially, within the context of recently popular research topic of oxygen redox cathode materials.

Author Response: Thanks for the helpful comments. We agree with the reviewer that Li_2MnO_3 itself may not be a high energy density electrode in the near future because of the irreversible degradation in the 1st cycle. However, Li_2MnO_3 is the key component of the $\text{Li}_2\text{MnO}_3\text{LiMO}_2$ layer-layer materials which have been as one of the most promising electrodes. Currently, they also suffer from some problems like the gradual voltage decay upon a number of cycles and attention has been paid to investigate the correlation between voltage decay and the potential Li_2MnO_3 degradation. Therefore, learning from how Li_2MnO_3 fails, can shed light on how to improve the derivated electrodes. Following the kind suggestion of the reviewer, we have added discussion into the Introduction as “Although pure Li_2MnO_3 is now viewed as an unrealistic cathode material for commercial Li ion battery applications due to its rapid degradation, the underlying mechanism of the failure is unclear. Here we reported experimental finding and theoretical modeling results which provides deeper insights on the underlying failure mechanisms.”

1. Ruther, R. E.; Samuthira Pandian, A.; Yan, P.; Weker, J. N.; Wang, C.; Nanda, J., Structural Transformations in High-Capacity $\text{Li}_2\text{Cu}_{0.5}\text{Ni}_{0.5}\text{O}_2$ Cathodes. *Chemistry of Materials* **2017**, 29 (7), 2997-3005.
2. Zheng, J.; Xu, P.; Gu, M.; Xiao, J.; Browning, N. D.; Yan, P.; Wang, C.; Zhang, J.-G., Structural and Chemical Evolution of Li- and Mn-Rich Layered Cathode Material. *Chemistry of Materials* **2015**, 27 (4), 1381-1390.
3. Zuo, Y.; Li, B.; Jiang, N.; Chu, W.; Zhang, H.; Zou, R.; Xia, D., A High-Capacity O₂-Type Li-Rich Cathode Material with a Single-Layer Li_2MnO_3 Superstructure. *Advanced Materials* **2018**, 30 (16), 1707255.
4. Lu, X.; Adkins, E. R.; He, Y.; Zhong, L.; Luo, L.; Mao, S. X.; Wang, C.-M.; Korgel, B. A., Germanium as a Sodium Ion Battery Material: In Situ TEM Reveals Fast Sodiation Kinetics with High Capacity. *Chemistry of Materials* **2016**, 28 (4), 1236-1242.

REVIEWERS' COMMENTS:

Reviewer #1 (Remarks to the Author):

In-situ TEM study of batteries is highly interesting to a lot of researchers in the field by allowing fast reaction kinetics to be seen at the atomic scale. Especially it can dynamically observe what happened in a battery, critically revealing genuinely new science discovery or failure mechanism. After the revision, the authors properly addressed the questions raised by the reviewers. The conclusions are reasonably supported by the evidence provided. I feel that the current version satisfies the high standard of Nature Communications.

Therefore, I suggest its publication in this journal !

Reviewer #2 (Remarks to the Author):

The revised manuscript has addressed most of the major concerns. One of them is the electron beam irradiation effect. Authors have carried out a controlled experiment to verify their results. I appreciate their efforts and recommend the paper for publication.

Reviewer #3 (Remarks to the Author):

After reviewing the changes made by the authors, I am mostly satisfied with the new version and the answers.

The answer on the operando TEM conditions could have been more precise (galvanostatic or not?). The TEM simulations are not so conclusive since they do not really explain the width of the stacking fault. A simulation of the fault could have been shown.

Finally the EELS data are really not good enough (or processed sufficiently) to really answer the question on the change of composition/electronic structure from bulk to the surface.

All other comments were well treated by the authors. I would prefer a proper TEM simulation and at least calculation of position dependent O/Mn ratios (from EELS) to be at ease with the final version.

If a another new version is not demanded by the other reviewers, my two wishes should not be considered as compulsory before acceptance.